# Concurrent Validity and Reliability of an Inertial Measurement Unit for the Assessment of Craniocervical Range of Motion in Subjects with Cerebral Palsy

**DOI:** 10.3390/diagnostics10020080

**Published:** 2020-02-01

**Authors:** Cristina Carmona-Pérez, Juan Luis Garrido-Castro, Francisco Torres Vidal, Sandra Alcaraz-Clariana, Lourdes García-Luque, Francisco Alburquerque-Sendín, Daiana Priscila Rodrigues-de-Souza

**Affiliations:** 1Centro de Recuperación Neurológica de Córdoba (CEDANE), 14005 Córdoba, Spain; mcarperes@yahoo.es; 2Doctoral Program in Biomedicine, University of Córdoba, 14004 Córdoba, Spain; m72alcls@uco.es (S.A.-C.); lgarcial05@hotmail.com (L.G.-L.); 3Department of Computer Science and Numerical Analysis, Rabanales Campus, University of Córdoba, 14071 Córdoba, Spain; cc0juanl@uco.es (J.L.G.-C.); frantorresvidal@gmail.com (F.T.V.); 4Maimonides Biomedical Research Institute of Cordoba (IMIBIC), 14004 Córdoba, Spain; 5Physiotherapy Section, Faculty of Medicine and Nursing, University of Córdoba, 14004 Córdoba, Spain; drodrigues@uco.es

**Keywords:** inertial sensors, pediatric neurological disease, kinematics

## Abstract

Objective: This study aimed to determine the validity and reliability of Inertial Measurement Units (IMUs) for the assessment of craniocervical range of motion (ROM) in patients with cerebral palsy (CP). Methods: twenty-three subjects with CP and 23 controls, aged between 4 and 14 years, were evaluated on two occasions, separated by 3 to 5 days. An IMU and a Cervical Range of Motion device (CROM) were used to assess craniocervical ROM in the three spatial planes. Validity was assessed by comparing IMU and CROM data using the Pearson correlation coefficient, the paired t-test and Bland–Altman plots. Intra-day and inter-day relative reliability were determined using the Intraclass Correlation Coefficient (ICC). The Standard Error of Measurement (SEM) and the Minimum Detectable Change at a 90% confidence level (MDC_90_) were obtained for absolute reliability. Results: High correlations were detected between methods in both groups on the sagittal and frontal planes (r > 0.9), although this was reduced in the case of the transverse plane. Bland–Altman plots indicated bias below 5º, although for the range of cervical rotation in the CP group, this was 8.2º. The distance between the limits of agreement was over 23.5º in both groups, except for the range of flexion-extension in the control group. ICCs were higher than 0.8 for both comparisons and groups, except for inter-day comparisons of rotational range in the CP group. Absolute reliability showed high variability, with most SEM below 8.5º, although with worse inter-day results, mainly in CP subjects, with the MDC_90_ of rotational range achieving more than 20º. Conclusions: IMU application is highly correlated with CROM for the assessment of craniocervical movement in CP and healthy subjects; however, both methods are not interchangeable. The IMU error of measurement can be considered clinically acceptable; however, caution should be taken when this is used as a reference measure for interventions.

## 1. Introduction

Cerebral palsy (CP) comprises a group of disorders affecting the development of movement and posture, causing activity limitations, and is attributed to a non-progressive damage to the developing brain during the fetal period or in the first years of life [1]. According to the Surveillance of Cerebral Palsy in Europe, CP affects between 1 to 3 per 1000 live births [2,3], with a prevalence of 3 to 4 cases per 1000 among school-age children in the US [4]. Currently, CP is recognized as being the most common cause of serious permanent physical disability in childhood, although the prospect of survival in children with severe disability has increased in recent years. Cerebral palsy is associated with sensory deficits, cognitive deficits, communication and motor disabilities, behavioral problems, seizure disorders, pain and secondary musculoskeletal problems, with spastic paresis being one of the most common forms of presentation [5,6], affecting the magnitude of movement and motor control [7,8], including the craniocervical region. Thus, head movement alterations can impair temporomandibular joint functions [9], and increase the risk of falls [10]. Furthermore, certain disorders affecting the senses can lead to unusual head movements and these alterations of the head movements can in turn further affect the senses [11,12]. In addition, it is suggested that the evaluation of motor disorders should not be centered only on posture, but also on the analysis of movement [13]. All of the above increases the need for valid and reliable methods to study cervical movement in patients with CP.

Most of the assessment methods in CP are based on subjective measures that classify motor participation based on functional abilities [14,15,16]; however, more advanced approaches are necessary in clinical settings and research [17]. Inertial Measurement Units (IMUs) have been known to benefit motion assessments due to their portability, ease-of-application, and low energy consumption, in contrast to other complex electromagnetic devices or video-based optoelectronic systems, which can only be used in laboratory settings [18]. In fact, IMUs represent a scientific advancement in the bio-healthcare sector, by measuring the kinematics of body segments, since these are adapted to each body region and use specific protocols that must be validated [18,19,20]. Good reliability results regarding optical motion capture have been described for the assessment of cervical and thoracolumbar range of motion (ROM) [21,22]. Their use in neurological diseases includes balance assessments in multiple sclerosis [23,24], Parkinsonian tremor [25,26], or range of motion (ROM) in stroke [27]. Nevertheless, further studies are necessary to confirm the clinical and predictive importance of measurements with IMUs [13,23]. Additionally, future research is required to support this validity with other tools [28] in pediatric pathologies [18,29]. To date, in children with CP, spasticity in lower limbs has been studied, obtaining satisfactory results in terms of precision and reliability, superior to other alternatives, such as goniometry [28], and gait analysis [30].

Thus, the aim of this study was to determine the clinimetric characteristics of IMU, in terms of validity and reliability, for the assessment of cervical ROM in patients with CP. In addition, we sought to establish error threshold values and minimum detectable difference with IMU in the assessment of the cervical spine in patients with CP, in order to determine clinical effect. We hypothesized that IMU would show good concurrent validity with cervical range of motion device (CROM) and that the determination of ROM using IMU would reveal good intra- and inter-day reliability.

## 2. Materials and Methods

### 2.1. Subjects

A clinical measurement study assessing validity and reliability was designed using a two-stage repeated measures design. Patients with CP were recruited using non-probabilistic sampling of consecutive cases from the private Neurological Recovery Center of Córdoba (CEDANE) and the Rehabilitation Service of the Reina Sofía University Hospital of Córdoba (Andalusian Health Service), in Spain. The inclusion criteria were: male and female subjects aged between 4 and 14 years old; diagnosed with CP and poor head control; with the necessary cognitive and behavioral skills required for understanding tasks and following simple instructions for active participation in the study; Gross Motor Function Classification System (GMFCS) levels I-IV; medically stable. In addition, to ensure active movement against gravity, all subjects had to achieve, at least, a level of 3 in the Manual Muscle Test of cervical muscles [31,32]. The exclusion criteria were: aggressive or self-injurious behavior; involuntary or uncontrollable movements of the head; orthopedic surgery at least 1 year before the evaluation or 6 months from the administration of botulinum toxin; anti-spasticity medications at the time of the assessment; severe tactile hypersensitivity that hinders body alignment; severe visual limitations; bone deformities or contractures that prevent assessment; history of uncontrolled pain; participation in another biomedical research (and/or patients in a period of exclusion).

Control subjects were also selected for this study. These were subjects with no neurological or other impairments, matched for gender and age (±2 years). They were recruited from the Hospital and the University, as well as via the researchers’ personal contacts.

The parents or caregivers of all study subjects gave their informed consent in accordance with the tenets of the Declaration of Helsinki for inclusion before they participated in the study. The protocol was approved by the Ethics Committee of Reina Sofía University Hospital (act nº270, reference 3680, 6 November 2017 approved).

The sample size required to test the concurrent validity between the IMU and CROM was based on a bilateral Pearson’s correlation coefficient, assuming an expected correlation of r ≥ 0.60, a level of significance of 5%, and 90% power. Thus, we determined that at least 21 subjects were necessary in each group. In addition, based on previous studies [33,34,35], and considering an intraclass correlation coefficient (ICC) of 0.8, an accuracy of 0.23 and a level of significance of 5%, the estimated sample should comprise, at least 22 subjects (Tamaño de la muestra 1.1^®^ software, Bogotá, Colombia). Due to the short follow-up period, no data loss was expected.

### 2.2. IMU Assessment

An IMU Shimmer3 ^®^ sensor (Dublin, Ireland) was located on the patient’s forehead, attached to the head using a flexible and adjustable strap (Figure 1A). Orientation in the three planes of movement was obtained by a sensor at 50 Hz, connected to an android mobile phone using iUCOTrack © (Córdoba, Spain) [21,22] a software program for the acquisition and processing of the raw data obtained by the sensor, producing kinematic results. The patient performed three movements in each of the three spatial planes (flexion and extension in the sagittal plane, left and right rotation in the transverse plane, left and right lateroflexion in the frontal plane), recording the maximum values of each movement. The ICC among the three repetitions of each movement was over 0.8 in all cases.

### 2.3. CROM Assessment

The Cervical Range of Motion (CROM 3 ^®^, Lindstrom, MN, USA) device was used for the goniometry assessment, together with the IMU. This device has three spheres (2 inclinometers and a compass) to determine the ROM in the three spatial planes (Figure 1B). Its validity and reliability have been proven in cervical functional assessments for all movements [36,37]. The CROM cannot be adapted to fit different head sizes. Thus, semi-rigid foams were used to adjust the CROM to the children’s heads and to prevent any movement. As the CROM was applied together with IMU, three repetitions of each movement were also performed, for which the ICC of the three repetitions was over 0.75 in all cases.

### 2.4. Muscle Tone Assessment

Due to the influence of spasticity in ROM, muscle tone was assessed for flexor, extensor, and sternocleidomastoid muscles of CP subjects, using the Modified Ashworth Scale (MAS) [38,39]. This scale is widely used and easy to administrate, with moderate to good reliability in CP [40].

The MAS scale is scored as follows:

0: No increase in muscle tone.

1: Slight increase in muscle tone, manifested by a catch and release, or by minimal resistance at the end of the range of motion when the affected part(s) is moved in flexion or extension.

1+: Slight increase in muscle tone, manifested by a catch, followed by minimal resistance throughout the remainder (less than half) of the ROM.

2: More marked increase in muscle tone through most of the ROM, but affected part(s) easily moved.

3: Considerable increase in muscle tone, passive movement difficult.

4: Affected part(s) rigid in flexion or extension.

### 2.5. Procedures

The general recommendations for assessments in this patient profile were applied, meaning that evaluation and treatment strategies must include relatives or caregivers who are functionally involved and part of the daily relationship (relatives/caregiver/child) [41,42].

The evaluations were performed in a quiet room, with no other people present besides the subject, assessors, and relatives/caregiver. All people stood behind the study subject, except for the assessor, who read the CROM values. A non-swivel chair was used, adapted to the anthropometric characteristics of each subject, who were seated in a standardized manner, and secured with straps when necessary. Specific instructions were given to the subject for the performance of each movement, as follows: for flexion, “first, tuck in your chin, then move your head forward and down as far as possible”; for extension, “first, raise your chin, then move your head backward, looking up as far as possible until limited by tightness or discomfort”; for rotation in each direction, “turn your head, gazing at an imaginary horizontal line on the wall, as far as possible”; for lateral flexion in each direction, “stare straight ahead and side-bend your neck by moving your ear toward your shoulder as far as possible”. To avoid thoracic movement, the instructions were, “do not move your shoulders or change the amount of pressure applied to the backrest of your chair” [37]. Manual stabilization was provided during each movement to avoid movements other than those requested and to control for any proprioceptive or other sensorimotor problems that could occur during the static posture or the performance of the movements, when necessary. To control for the appearance of resistance to movement due to spasticity, an assessor performed stretches of the muscle, repositioning the joint in the position where the resistance appeared. Subsequently, a second examiner annotated the CROM values [43].

The Wong–Baker facial pain scale [44] was applied to assess whether patients suffered from pain throughout the evaluations. Its results were applied to interrupt the patient’s participation in the study.

The two movements in each spatial plane were added to obtain the ROM in each plane (flexion-extension range: flexion plus extension; rotational range: right rotation plus left rotation; side-bending range: right lateral flexion plus left lateral flexion). The use of the ROM in each plane has been described as an advantage to assess cervical movement due the possible discrepancies in determining the neutral position when half movements are assessed [45].

Data were collected on two different occasions, separated 3 to 5 days. On the first day, measurements were applied twice, separated by 5 min, without changing the position of the subject. On the first evaluation, both IMU and CROM were applied, to compare results between both devices, and on the second evaluation only IMU was used, for intra-day reliability purposes. The IMU assessment was repeated 3–5 days later, to analyze inter-day reliability (Figure 1C,D).

The assessor was blinded to the previous measures at the time of the new trial [18]. All intra-day and inter-day tests were performed by the same assessor, a physiotherapist with more than 15 years of experience in the evaluation of patients with CP.

### 2.6. Statistical Analysis

Frequencies and percentages were used to describe categorical variables. The arithmetic mean, standard deviation and 95% confidence intervals (95% CI) were used for quantitative variables, once normality and homoscedasticity were tested (Shapiro-Wilk and Levene’s tests, *p* > 0.05).

Spearman’s rho correlation coefficient (r_s_) was used to identify associations between cervical muscle tone and ROM, assessed with the CROM and IMU. Correlation coefficient values were considered poor when values were below 0.20, fair for values between 0.21 to 0.50, moderate from 0.51 to 0.70, very strong from 0.71 to 0.90, and almost perfect from 0.91 to 1.00 [46].To identify possible specific characteristics in craniocervical ROM of CP, IMU, and CROM data from each assessment and spatial plane were compared between groups by unpaired t-tests.

#### 2.6.1. Concurrent Validity Analysis

To assess concurrent validity, the Pearson’s correlation coefficient (r) was applied for data obtained by IMU and CROM when applied together, that is, during the first assessment on the first day, with the same interpretation based on the Spearman rho [46]. The paired t-test was also used to analyze the differences between the means of both methods in each spatial plane ROM. In addition, Bland–Altman plots were constructed for each ROM [47,48]. The mean bias, defined as the average of the differences between both methods of measurement, was determined, together with limits of agreement (LoA), providing an estimate of the interval where 95% of the differences between both methods lie, and defined as the bias ±1.96 standard deviations of differences.

#### 2.6.2. Reliability Analysis

The relative reliability of the measurements of each ROM evaluated with the IMU was determined by calculating ICC for intra-day and inter-day reliability (ICC2,1) [49]. The intra-day reliability was calculated based on the assessments performed on the first day, and the inter-day reliability was estimated between the first assessment on the first day and the assessment performed on the second day. For all analyzes, ICC values were considered poor when values were below 0.20, reasonable from 0.21 to 0.40, moderate from 0.41 to 0.60, good from 0.61 to 0.80, and very good from 0.81 to 1.00 [34].

The absolute reliability was determined by calculating the SEM and the Minimum Detectable Change at 90% confidence level (MDC_90_) for each movement:SEM = SD_pooled_ × √ 1 − ICC,
where SD_pooled_ is the standard deviation of the scores from all subjects;
MDC_90_ = SEM × √2 × 1.64.

The SEM provides a value for the random measurement error in the same unit as the measurement itself quantifies the variability within the subject and reflects the amount of measurement error for any given test (intra-day reliability) and for any test occasion (inter-day reliability) [50,51]. The MDC is an estimate of the smallest amount of change between separate measures that can be objectively detected as a true change outside the measurement error [50,52], and the MDC_90_ is frequently used to identify the effectiveness of an intervention [33].

For a better control of type I error risk, due to the repeated comparison among CROM and IMU data, a two-way ANOVA, with Evaluation (CROM; IMU first assessment on the first day; IMU second assessment on the first day; IMU assessment on the second day) as the within-subject factor, and Group (CP group; control group) as the between-subjects factor, was performed for each spatial plane ROM. The evaluation-by-group interaction and both factors were of interest. Should the interaction or any of both factors reveal significance, the Bonferroni’s post-hoc test was used to verify whether a difference existed between the groups and/or within groups (view Appendix A).

All hypothesis tests were bilateral and considered significant if *p* was less than 0.05. The data were managed and analyzed with IBM-SPSS^®^, version 25 (Armonk, NY, USA).

## 3. Results

The present study consisted of 46 participants (CP group: *n* = 23; Control group: *n* = 23), 61% of whom (*n* = 28) were female. Their average age was 8.9 years with a standard deviation of 3.2 years. The GMFCS showed that 47.8% of the CP subjects were classified as level I, 17.4% as level II, 4.4% as level III, and 30.4% as level IV. No patient showed a value of 2 or more in any muscle and over 30% of CP subjects had no impairment in muscle tone, according to the MAS. This means that muscle tone suffered, at most, a slight increase in the CP subjects. No study subjects suffered pain or other difficulties when undergoing the complete evaluation. Other basic descriptive characteristics of the groups are given in Table 1.

The correlation analysis between MAS and ROM, assessed by the CROM and the IMU, showed a common trend, with flexor, extensor, and sternocleidomastoid muscles of CP subjects significantly and negatively correlated with rotational ROM (in all cases: the higher the muscle tone, the lower the ROM). Thus, the tone of flexor and extensor muscles correlated with: CROM: r_s_ = −0.504; IMU first assessment on the first day: r_s_ = −0.510; IMU second assessment on the first day: r_s_ = −0.483; IMU assessment on the second day: r_s_ = −0.412. Right and left sternocleidomastoid muscle tone correlated with: CROM: r_s_ = −0.433; IMU first assessment on the first day: r_s_ = −0.437; IMU second assessment on the first day: r_s_ = −0.420; IMU assessment on the second day: r_s_ = −0.410. No correlation was identified in the planes of flexion-extension and side-bending.

No differences were detected between CP subjects and controls in each ROM for any of the assessments, regardless of the method of measurement (*p* > 0.05).

Additionally, as reported in Appendix A, the two-way ANOVA of the ROM of the three spatial planes showed a consistent pattern, with neither evaluation-by-group interaction nor Group factor significance, although the Evaluation factor detected statistical differences (*p* ≤ 0.02). The post-hoc analysis of the Evaluation factor showed differences between CROM and the IMU assessments, with no differences among the three IMU assessments. The only exception to this pattern was the post-hoc analysis of the Evaluation factor concerning the rotational plane ROM, with statistical differences, exclusively, between the CROM and IMU assessments on the second day.

### 3.1. Concurrent Validity

The measurements obtained by the first IMU assessment on the first day correlated highly with the measurements of the CROM for flexion-extension and side-bending ranges in both groups (r > 0.9), although rotation range correlations were smaller (0.6 < r < 0.8). Significant differences between both methods were observed in all ROMs, with the exception of the rotational range in the control group (Table 2).

The Bland–Altman plots (Figure 2 and Figure 3) indicated bias below 5º between both measurement systems for craniocervical ROMs, except for the rotation range of the CP group (mean bias: 8.2º). Nevertheless, the distance between LoAs for all ROMs and both groups were over 23.5º, with the exception of flexion-extension range in the control group (distance between LoAs: 12.1º). Finally, some outliers were found on the Bland–Altman plots of the CP group.

### 3.2. Intra-Day and Inter-Day Reliability

Table 3 shows absolute reliability results. For the intra-day reliability, all ICCs for both groups were from 0.82 to 0.93, with the 95% CI showing a common trend of (upper limit: ICC+0.2, lower limit: ICC−0.2). Nevertheless, absolute reliability data were variable, although all SEM were below 8,5º, and MDC_90_ between 11.4º (side-bending range of the control group), and 19.4º (rotational range of CP group).

For the inter-day reliability, all ICC values were higher than 0.8 and the 95% CI also showed a trend of (ICC+0.2; ICC−0.2), except for rotational range of the CP group (ICC = 0.53) with a wide 95% CI. SEM showed higher values compared to the intra-day values in all ROMs and both groups. Furthermore, the SEM and MDC_90_ were higher than 20º for flexion-extension and rotational ranges in CP subjects, and for flexion-extension range in controls. 

## 4. Discussion

This is the first methodological study to assess the validity and reliability of IMUs for the assessment of craniocervical ROM in CP. SEM and MDC values were also provided for possible applications in clinical settings. According to the results, the hypotheses were partially confirmed. Thus, high correlations were found between the IMU and the CROM, although there were statistical differences when data from both methods were compared, and the range between LoAs was high. In summary, the IMU showed good concurrent validity regarding CROM; however, the methods were not interchangeable. In addition, intra-day and inter-day reliability were, in general, very good; however, the SEM and MDC were too high for inter-day comparisons for both CP and healthy subjects, hampering their application in clinical settings.

Limited information is available in the literature regarding the validity and reliability of portable devices for the measurement of cervical ROM in CP [45], which hampers comparisons with other studies. According to previous research, the validity and reliability of IMUs depend on the specific context where they are applied [18]. Thus, our results show an almost perfect correlation between the IMU and CROM, except in the case of the rotational plane, and a more questionable agreement. The exception found for the rotational plane may be due to the location of the transverse plane, where rotational movements are included, and which is perpendicular to the sagittal and frontal planes, where flexion-extension and side-bending movements are included, respectively. Flexion, extension, right lateral flexion, and left lateral flexion of the cervical spine, when originated from the neutral position, are performed in the direction of the force of gravity, making these easier to perform and, consequently, easier to reproduce in a homogenous fashion, compared to rotations, which require a continuous balance against gravity, an action which is compromised in CP [13]. Indeed, the lower validity of the results concerning the rotational plane has been consistently reported in healthy adult subjects [53], thus highlighting the significant challenge related to the measurement of ROM in rotational plane movements.

The validation pattern found in this study agrees with a former report by Chang et al. [54], who identified high correlations, although differences in craniocervical ROM values were reported, specifically for side-bending to the right and rotation to the left, when an electromagnetic portable device is compared to a universal goniometer. Although some methodological differences can be described between the research by Chang et al. and our own, such as the use of a sample comprising only healthy subjects, the assessment of movements which was performed from the neutral position, the use of a universal goniometer, and the absence of inter-day assessments, the interpretation of the agreement between both methods were the same. Thus, although the range between LoAs only defines the intervals of agreements, and not whether those limits are acceptable or not [48], we agree with Chang et al. that LoAs over 12º impaired the interchangeability of measurement methods [54]. Indeed, the assessment of craniocervical motion poses a greater challenge, compared to the motion of peripheral joints, for several reasons. First, because multiple joints are involved in craniocervical mobility. Second, due to the difficulty of avoiding thoracic spine movements, which can significantly modify the magnitude of the movements. Third, craniocervical motion is three-dimensional and movement in one axis (primary movement) can be influenced by those in other spatial planes (coupled movements) [54], which can vary among individuals [55] and can be altered by the presence of diseases [56]. All these circumstances may have influenced our results, as CP is commonly associated with a loss of cervical motor control [11,12].

Analogic goniometry using CROM has been previously established as the method of reference for evaluating neck motion [36,37], however, recently, other 3-D kinematic devices have also been proposed [57], and some circumstances support the use of digital devices to assess neck motion, such as the need of one less assessor to obtain the CROM data, a proper adjustment to the shape and size of the head without the need for additional elements (i.e., semi-rigid foams), for use in the pediatric population, and the elimination of the reading error associated with analogic devices [54,58]. This is in agreement with Paulis et al. [27], who support the objective and automatized collection of IMUs data to assess ROM in elbow muscle spasticity after stroke. Furthermore, a recent systematic review has suggested that rehabilitation research and health care services could benefit from IMUs because they provide valid data to assess ROM and joint orientation [53].

Our study showed good to very good relative reliability for intra-day and inter-day comparisons and no differences among IMU assessments in each group. It is known that the ICC increases with larger between-subjects variance [52]. In fact, we found a high variability of the data, with standard deviations over 15º in almost all cases, which means approximately 20% of the mean values of some ROMs, independent of the clinical condition. It has been described that cervical ROM shows an important dependence on age [45], which could explain the variability of the results. These interpretations of ICCs are in consonance with previous studies using IMUs in neurological diseases [27]. The exception to the high ICCs was the inter-day reliability of rotational range in the CP group, as occurred with the validity assessment. Again, it may be more difficult to repeatedly reproduce cervical rotations in a homogenous manner, as opposed to other movements, due to the balance deficiencies of subjects with CP [13]. As commented, for validity purposes, the poorer results of the rotational plane have been also found in healthy adult subjects, including ICC values below 0.8 [53]. Further research and innovative assessment approaches are necessary to improve the quality of rotational plane ROM measurements.

The SEM and MDC were acceptable for intra-day reliability, although greater for inter-day reliability, which makes their clinical applicability difficult. Thus, it is difficult to achieve an effect of more than 20º when a therapeutic intervention is applied in research or clinical settings, at least, for comparisons between different days. The previously commented high variability of the data may explain this low absolute reliability, which has been previously identified for walking performance and physical activity in CP and healthy subjects [59]. Indeed, although most studies show that the measurement error of the IMUs for motion assessment is between the 2º and 5º [18,54], the SEMs of this study were all over 5º, which can be considered clinically acceptable, according to previous studies, both in neurological patients [27] and healthy subjects [53], at least for intra-day comparisons. Furthermore, specifically for inter-day calculations, two more sources of variability may explain these results. First, spasticity varies from one movement to another, and even more when the assessments are performed on different days [60], making it difficult to ensure that the evaluations of the CP subjects were performed in the same clinical conditions. Furthermore, it is known that spasticity can be influenced by apprehension, excitement, and the position in which the child is assessed [61], which can increase the variability of the ROM results, mainly in inter-day evaluations. Second, the magnitude of a training effect or compensation cannot be calculated due to the repetitions [54], however, all the mean ROMs on the second day were higher than those of the first day in both groups, which may have also influenced the absolute reliability between days. The possible changes affecting the exact placement of the IMU between the two assessment days may also partially explain the worse absolute reliability for the inter-day comparisons [27]. Finally, although the variability is supposed to be small, calibration may be necessary for each evaluation to ensure the proper function of the gyroscope and magnetometer [54]. Previous research has identified that calibration in certain specific populations may be more challenging, such as CP patients [53].

No pain was experienced by the study subjects during the procedures, and no assessment was interrupted due to the evaluation protocol. This means that the application of IMUs for craniocervical ROM assessment is tolerable, safe, and innocuous when applied to CP and healthy children. Although the body mass index (BMI) showed differences between groups, we believe that this does not influence the study results, due to the simplicity of the task performed. Furthermore, CP subjects revealed increased BMI values compared to healthy subjects [62,63], which is a common health problem in this population.

Surprisingly, no differences in ROM were detected between CP subjects and controls, although most CP clinical presentations are associated with spasticity. However, the level of increased cervical muscle tone in the study sample can be considered as being low, which is a plausible explanation of these results. Indeed, regardless of the method of measurement, only fair to moderate correlations were found, exclusively between the tone of cervical muscles and rotational ROM, perhaps due to the fact that greater motor control is necessary to perform rotations, as previously described, with no correlations in any other spatial plane. Furthermore, the association between spasticity, hypertonia, and ROM is not completely understood at this time [60]. On the contrary, the reduction of craniocervical ROM has been described as a characteristic of several musculoskeletal and neurological diseases. Thus, cervicogenic headache in children determines reduced flexion, extension, and lateroflexion, although not rotational movements [64], plagiocephaly limits cervical ROM, especially in the rotational plane [65], and congenital muscular torticollis reduces ROM in frontal and transversal planes [66]. In conclusion, specific craniocervical ROM is not a characteristic of CP in children, at least when muscle tone is slightly increased.

It has been suggested that subjects with motor disorders could benefit from IMUs for the following three purposes: (1) Objective quantification of motor disorders; (2) Proprioceptive enhancement through visual-motor feedback; (3) Functional compensation via an inertial person-machine interface [29]. From a clinical assessment point of view for CP, IMUs have been successfully applied for the stimulation and analysis of activity using interactive games [67], for the assessment of lower limb spasticity [58], during gait [30], and for the assessment of specific characteristics in the cervical spine in small samples [11,68]. Following the increasing interest and evidence of the benefits of IMU application in pathological populations, in terms of guiding clinical decision making (e.g., quantify deficits and determine progress in time) [69], the current study adds the assessment of cervical ROM to the field of research of IMU in CP.

Despite the promising results of the current study, some limitations were identified. The applicability of findings is limited to similar samples and assessment protocols. A wider scope is necessary to establish conclusions regarding specific GMFCS levels or other age ranges. Furthermore, the current study only assessed the ROM of simple movements in a specific and controlled setting, which limits the applicability of the results to more complex tasks and day-to-day conditions [18]. In fact, although more simple movements are used to produce better clinimetric properties [70], this approach did not solve the common measurement problems of rotational plane mobility [53]. The sample size was relatively small, and several variables showed a high variability, which could have affected the strength of the comparisons. No inter-assessor reliability was evaluated, although the automatized process with IMUs makes an inter-assessor error difficult, as commented. Finally, some previous research has recommended the use of two IMUs to assess cervical ROM [45,71], but we preferred the application of one IMU adding a manual stabilization during each movement to avoid unwanted body motions, due to the difficulties to maintain a thoracic sensor fixed in children, and the need of an additional support on trunk in some CP subjects. Further research is necessary, considering additional factors, such as other movement characteristics, including velocity, acceleration or coupled angles, and innovative assessment protocols, with a special focus on complex and day-to-day tasks and rotational plane movements, and larger sample sizes, in order to standardize technical procedures and obtain accurate and normative data [53].

## 5. Conclusions

A high correlation was found between IMU and CROM for the assessment of craniocervical motion among individuals with CP and healthy subjects. However, both methods are not interchangeable. In CP subjects, the error of measurement in IMU can be considered clinically acceptable for the sagittal and frontal planes, although not for the transverse plane. When used as a reference measure for interventions, neck ROMs must achieve very high changes to ensure that the detected changes are significant. Future studies should be conducted to establish the normative data of craniocervical ROMs for specific population subgroups.

## Figures and Tables

**Figure 1 diagnostics-10-00080-f001:**
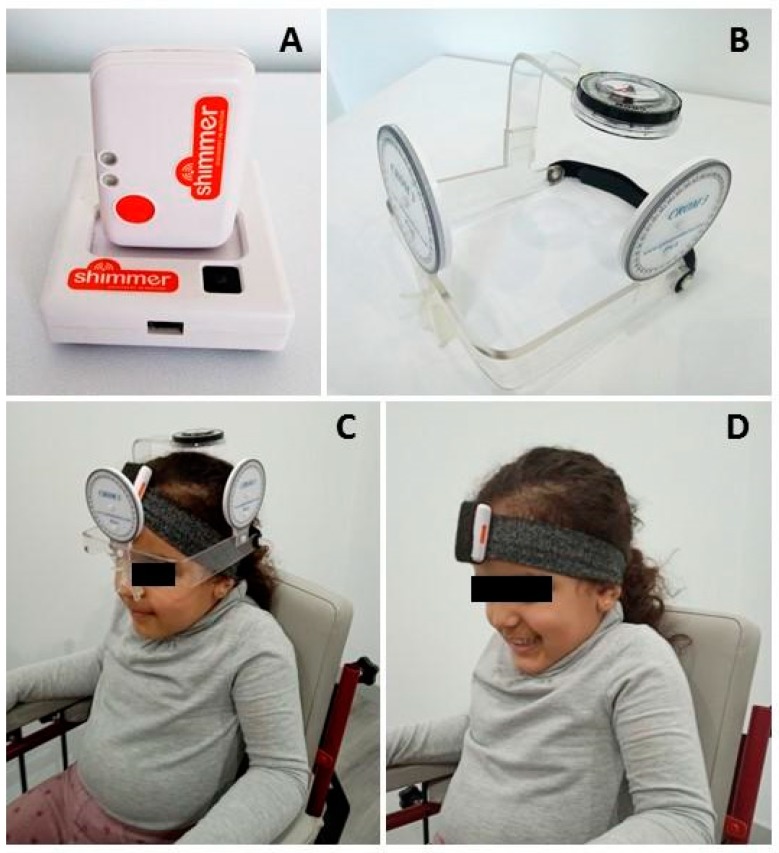
Devices and procedure of assessment. (**A**) Inertial Measurement Unit (IMU) Shimmer3 ^®^ (Dublin, Ireland); (**B**) Cervical Range of Motion (CROM) 3 ^®^ device (Lindstrom, MN, USA); (**C**) positioning of IMU and CROM to assess craniocervical range of motion (first assessment on the first day); (**D**) positioning of IMU to assess craniocervical range of motion (second assessment on the first day, and assessment on the second day).

**Figure 2 diagnostics-10-00080-f002:**
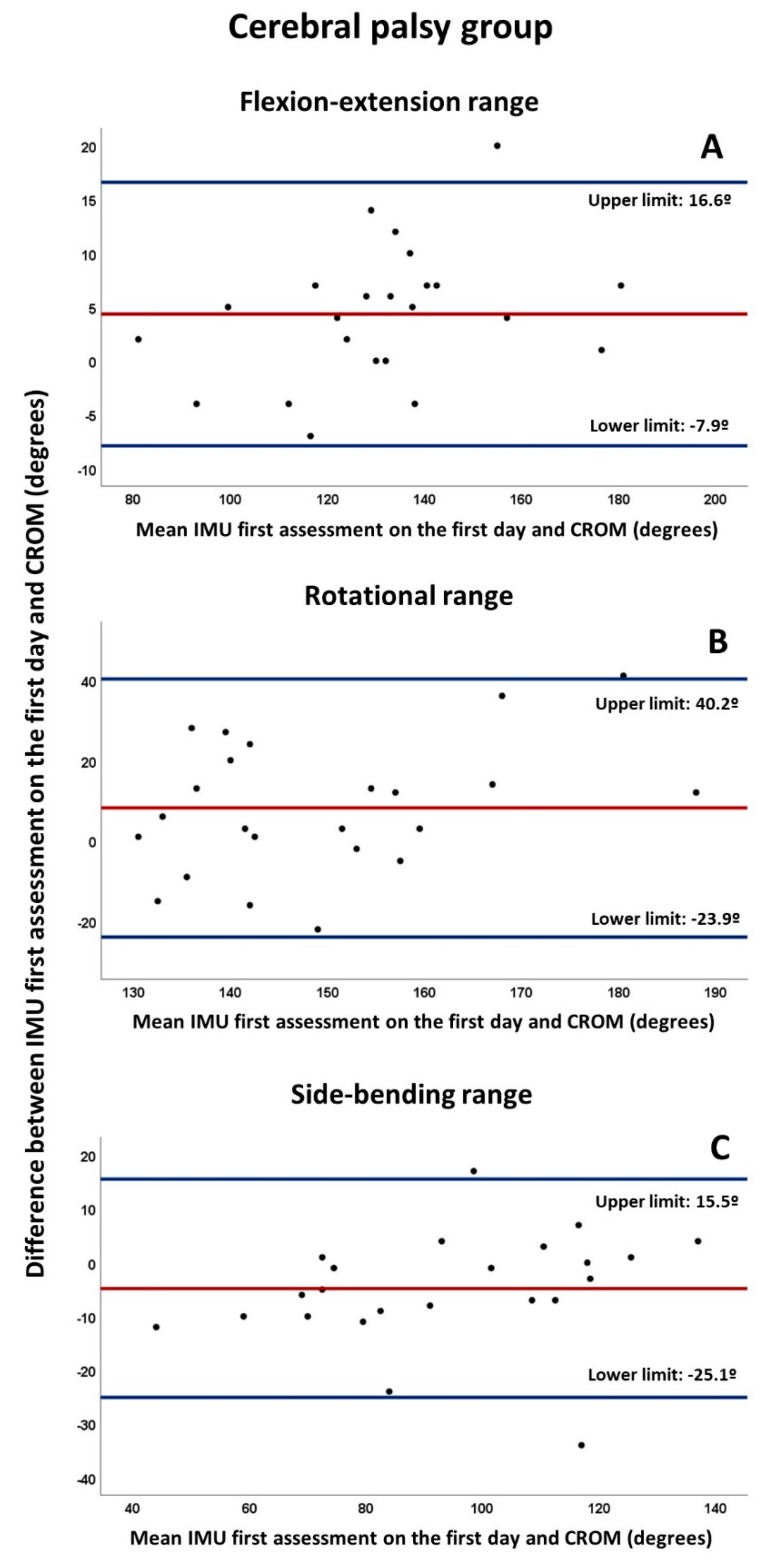
Bland–Altman plots for craniocervical ranges of CP group measured by IMU first assessment on the first day and the CROM in (**A**) flexion-extension range, (**B**) rotational range, (**C**) side-bending range. The red line indicates the mean bias, whereas the blue lines refers to its upper and lower limits (mean ± 1.96 standard deviation). All plots were developed as follows: the Y axis corresponds to the differences between the paired values of both methods (IMU-CROM), whereas the X axis represents the respective value of the average of both (IMU + CROM/2).

**Figure 3 diagnostics-10-00080-f003:**
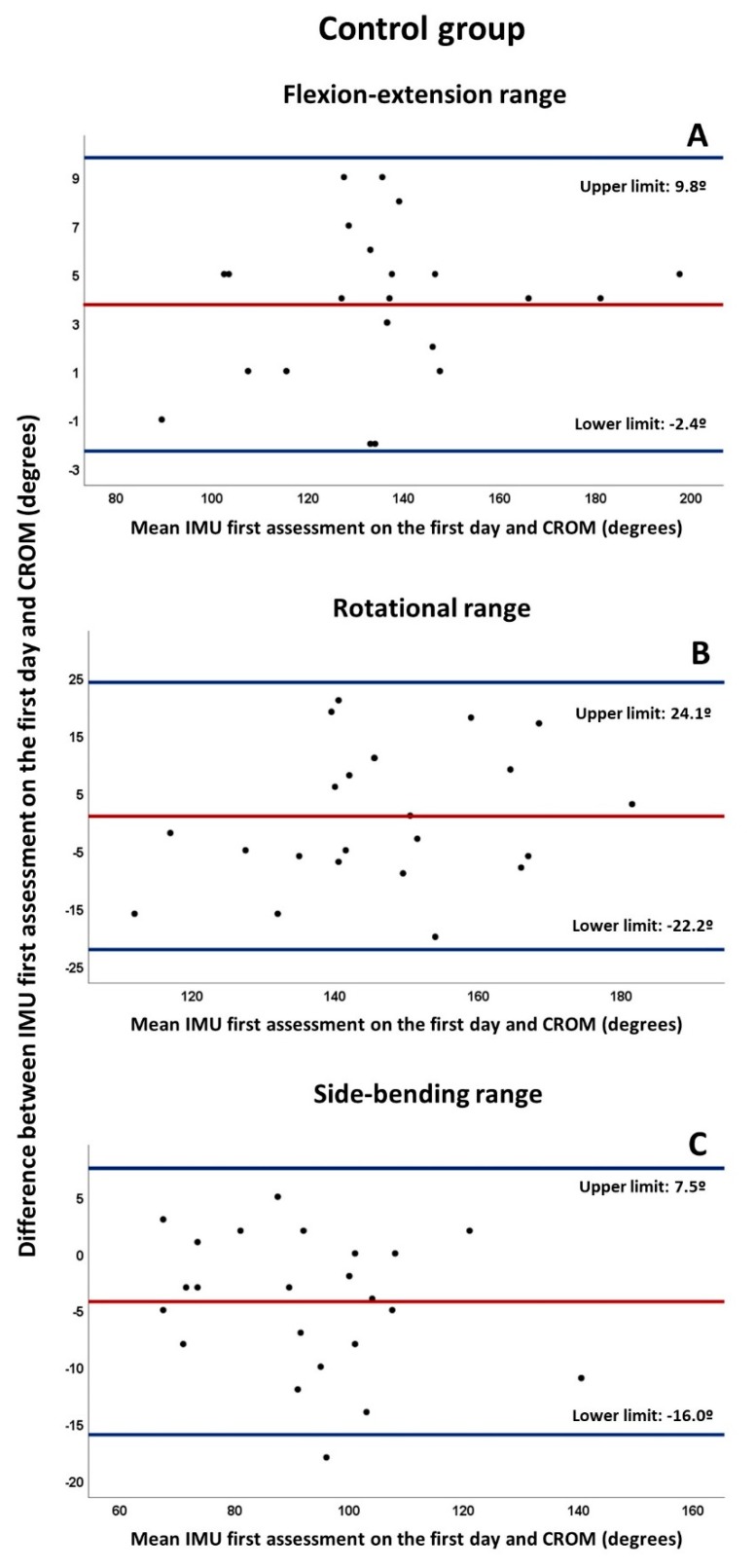
Bland–Altman plots for craniocervical ranges of the Control group measured by IMU first assessment on the first day and the CROM in (**A**) flexion-extension range, (**B**) rotational range, (**C**) side-bending range. The red line indicates the mean bias, whereas the blue lines refer to its upper and lower limits (mean ± 1.96 standard deviation). All plots were developed as follows: the Y axis corresponds to the differences between the paired values of both methods (IMU-CROM), whereas the X axis represents the respective value of the average of both (IMU + CROM/2).

**Table 1 diagnostics-10-00080-t001:** Demographics and clinical characteristics of the subjects.

	CP Group (*n* = 23)	Control Group (*n* = 23)	*p*-Value
**Age (years)**	9.2 (3.2)	8.7 (3.3)	0.594
**Sex (women/men)**	14/9	14/9	
**Weight (kg)**	28.3 (12.7)	34.6 (16.8)	0.161
**Height (m)**	1.32 (0.20)	1.36 (0.20)	0.503
**BMI (kg/m^2^)**	15.5 (3.4)	17.5 (3.5)	0.049*
**GMFCS level (frequency)**	**I**: 11; **II**: 4; **III**: 1; **IV**: 7	-	-
**Flexor muscles tone level (frequency)**	**0**: 10; **1**: 7; **1+**: 6	-	-
**Extensor muscles tone level (frequency)**	**0**: 10; **1**: 7; **1+**: 6	-	-
**Right sternocleidomastoid muscles tone level (frequency)**	**0**: 9; **1**: 8; **1+**: 6	-	-
**Left sternocleidomastoid muscles tone level (frequency)**	**0**: 9; **1**: 8; **1+**: 6	-	-

Quantitative data are expressed as mean (standard deviation). Abbreviations: GMFCS, Gross Motor Function Classification System; BMI, body mass index. * indicates *p* < 0.05.

**Table 2 diagnostics-10-00080-t002:** Concurrent validity between the first IMU assessment performed on the first day of measurements and CROM by groups.

Spatial Plane	IMU First Assessment on the First Day (Standard Deviation)	CROM Assessment (Standard Deviation)	Pearson r (*p*-Value)	Student’s t-test (*p*-Value)
**CP group (*n* = 23)**				
**Flexion-Extension**	133.3 (24.6)	129.0 (22.4)	0.969 (<0.001)	−3.333 (0.003)
**Rotational**	153.5 (19.9)	145.3 (14.7)	0.601 (0.003)	−2.396 (0.026)
**Side-bending**	91.3 (25.7)	96.1 (23.3)	0.916 (<0.001)	2.236 (0.036)
**Control group (*n* = 23)**				
**Flexion-Extension**	137.0 (24.5)	133.3 (23.9)	0.992 (<0.001)	−5.771 (<0.001)
**Rotational**	147.0 (19.0)	146.1 (16.1)	0.786 (<0.001)	−0.371 (0.714)
**Side-bending**	90.7 (17.5)	94.9 (18.9)	0.949 (<0.001)	3.413 (0.002)

Abbreviations: IMU, Inertial Measurement Unit; CROM, Cervical Range of Motion; CI, confidence interval; CP, cerebral palsy. Evaluation data are expressed in degrees.

**Table 3 diagnostics-10-00080-t003:** Intra-day and inter-day reliability of the IMU by groups.

		**Intra-Day Reliability**		
**Spatial Plane**	**IMU Second Assessment on the First Day** **(Standard Deviation)**	**ICC (95%CI)**	**SEM (º)**	**MDC_90_ (º)**
**CP group (*n* = 23)**				
**Flexion-Extension**	138.8 (26.5)	0.900 (0.762, 0.958)	8.0	18.6
**Rotational**	151.3 (20.1)	0.821 (0.600, 0.920)	8.3	19.4
**Side-bending**	91.3 (23.7)	0.925 (0.822, 0.968)	6.7	15.5
**Control group (*n* = 23)**				
**Flexion-Extension**	135.0 (24.7)	0.893 (0.750, 0.955)	7.9	18.4
**Rotational**	147.6 (20.3)	0.902 (0.750, 0.961)	5.4	12.6
**Side-bending**	86.3 (16.1)	0.913 (0.772, 0.965)	5.0	11.4
		**Inter-day reliability**		
	**IMU assessment on the second day** **(standard deviation)**			
**CP group (*n* = 23)**				
**Flexion-Extension**	140.8 (26.4)	0.873 (0.680, 0.947)	9.0	21.1
**Rotational**	156.2 (19.5)	0.533 (0.117, 0.803)	13.3	30.9
**Side-bending**	94.4 (24.8)	0.890 (0.743, 0.953)	8.3	19.3
**Control group (*n* = 23)**				
**Flexion-Extension**	140.2 (29.5)	0.831 (0.602, 0.928)	11.0	25.6
**Rotational**	153.2 (19.1)	0.846 (0.601, 0.935)	6.5	15.1
**Side-bending**	93.1 (16.5)	0.864 (0.653, 0.946)	6.1	14.2

Abbreviations: IMU, Inertial Measurement Unit; ICC, Intraclass Correlation Coefficient; CI, confidence interval; SEM, Standard Error of Measurement; MDC, Minimum Detectable Change. Evaluation data are expressed in degrees.

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
