# Peer review of "Concurrent Validity and Reliability of an Inertial Measurement Unit for the Assessment of Craniocervical Range of Motion in Subjects with Cerebral Palsy"

_diagnostics, 2020, doi:10.3390/diagnostics10020080_

Round 1

Reviewer 1 Report

General comment:

The paper presents a method for evaluating the cervical range of motion of patients with cerebral palsy aiming to complete the arsenal of devices allowing a better clinical follow up of those patients. It compares the results of the new device (IMU) with the gold standard (CROM). While the scientific question is convincingly highlighted and well referenced, some aspects of the methodology are not sufficiently developed and seem inappropriate as well as the research design. These are the main shortcomings of the study that need major revision.

Specific comments:

1. Introduction:

The introduction provides sufficient information of the working domain. References are relevant with respect to the topic.

2. Materials and methods:

The studied population is clearly defined. Ethical issues are well considered. Justification of samples sizes is provided.

Concerning exclusion criteria it would be useful to consider (i) the history of uncontrolled pain (since pain is a common feature in CP) and (ii) the participation to another biomedical research (and/or patients in a period of exclusion).

In the Procedures chapter, the authors recommend in line 160 "do not move your shoulders or change the amount of pressure applied to the backrest of your chair”. However, it is difficult to be sure about this since PC patients have proprioceptive and other sensorimotor problems. Authors must provide a comment in this sense.

One of the main issues of the methodological approach is the absence of measurements through CROM in the three measurement times (CROM only used during the first experimentation). Since CROM is the method of reference and its reliability has been proved, why the authors did not perform CROM measurements in the two other times ? If the measurements were done, they have to be included. This aspect is essential since it defines the statistical method to be used for comparing both methods (IMU versus CROM).

In this case, the use of a t-test comparison is not appropriate. Why the authors did not use ANOVA (each patient or line representing a repeated measure) followed by a multiple comparison post-test ? If an unpaired t-test is used when comparing more than two groups, the p value obtained has to be divided by the number of groups for the obtention of a correct statistical threshold which is not the case in the current study.

3. Results: 

Some results are rather unusual. although the correlation between the two methods for measurement is particularly high (flexion-extension > 0.9 and side-bending > 0.9 for both groups), the values are significantly different when comparing both methods. This is not logical when, in addition to that, the SD represent almost 20% of the values. This issue questions the reliability and pertinence of the Student test for statistically comparing both groups (see my previous comment concerning the multiple comparison methods and the correction of statistical thresholds)?

Table 2 is rather unclear and could be simplified (e.g. Student-t value and exact p value and * are redundant; mean difference and 95%CI are difficult to read, what is the usefulness of these details?)

"LoAs" is not sufficiently explained nor declared in the methodology. It's relevance is not sufficiently commented.

Figures 2 and 3 are difficult to interpret. It is said that craniocervical ranges are represented for IMU and CROM but few is explained inside the figures. Please explain better in the legend the meaning of these graphical representations.

Concerning Intra-day and Inter-day Reliability, it is difficult to explain why the authors do not use as a control the CROM method, which is been proved to be reliable, in particular when comparisons between the two measurement methods at initial time showed a significant difference. In this sense, what is the rationale of the intra-day and inter-day comparisons solely through IMUs ?

Since the reliability of IMUs is only partially validated in the first part of the study, the methodology for the second part is not correct and should include CROM measurements. This will allow to use an adapted statistical method which will allow to provide the right statistical results for two-by-two comparisons (ANOVA for repeated measures followed by a multiple comparison post-test; see my previous comments).

Discussion:

Since some methodological aspects are inadequately considered, the authors cannot affirm that the IMUs have a good concurrent validity regarding CROM and a very good intra-day and inter-day reliability (lines 266-274).

In line 285-286, the authors argue that the low correlation of CROM an IMU found in the rotational plane is due to the postural impairment of CP patients. This can be overcome by using a moulded sit and back braces for example. A first step is to prove the interchangeability of both methods (CROM and IMU) and the reliability of IMU in optimal experimental conditions (i.e. without major biais) and then its use in day-to-day conditions (e.g. in a CP patient seated in a conventional chair).

Author Response

Response to Reviewer 1 Comments

General comment:

Point 1: The paper presents a method for evaluating the cervical range of motion of patients with cerebral palsy aiming to complete the arsenal of devices allowing a better clinical follow up of those patients. It compares the results of the new device (IMU) with the gold standard (CROM). While the scientific question is convincingly highlighted and well referenced, some aspects of the methodology are not sufficiently developed and seem inappropriate as well as the research design. These are the main shortcomings of the study that need major revision.

Response 1: The authors appreciate the reviewer’s comments and the opportunity to enhance our manuscript. We thank the reviewer for their thorough and constructive comments on our previous submission. In preparing this revised manuscript, we have carefully reviewed the methodology, further describing the assessments and procedures. In addition, we have provided further details about the research design and how it was implemented. The Discussion was partially rewritten according to the reviewer’s comments.

We have marked all changes in the new version (file: diagnostics-694750_R1-WithFigures&Tables) of the manuscript in red text.

Specific comments:

Introduction:

Point 2: The introduction provides sufficient information of the working domain. References are relevant with respect to the topic.

Response 2: The authors appreciate this positive comment.

Materials and methods:

Point 3: The studied population is clearly defined. Ethical issues are well considered. Justification of samples sizes is provided.

Response 3: The authors appreciate this comment.

Point 4: Concerning exclusion criteria it would be useful to consider (i) the history of uncontrolled pain (since pain is a common feature in CP) and (ii) the participation to another biomedical research (and/or patients in a period of exclusion).

Response 4: We thank the reviewer for noticing this omission. Indeed, no subjects with pain or included in other research participated in the current study. We have now added both exclusion criteria, according of the suggestion.

Lines 101-102: history of uncontrolled pain; participation in another biomedical research (and/or patients in a period of exclusion).

Point 5: In the Procedures chapter, the authors recommend in line 160 "do not move your shoulders or change the amount of pressure applied to the backrest of your chair”. However, it is difficult to be sure about this since PC patients have proprioceptive and other sensorimotor problems. Authors must provide a comment in this sense.

Response 5: We applied a previously described protocol that found good reliability for CROM measurements in subjects with and without neck pain patients. In this protocol, researchers gave specific instructions for the performance of each movement (Fletcher & Bandy, 2008). To avoid any uncontrolled movement or other kind of problems, such as the appearance of spasticity, an assessor provided manual stabilization and oversaw the procedure, controlling any problems that could arise.

We have rewritten the sentence to improve the understanding of the protocol.

Lines 169-172: Manual stabilization was provided during each movement to avoid movements others than those requested and to control for any proprioceptive or other sensorimotor problems that could occur during the static posture or the performance of the movements, when necessary.

Point 6: One of the main issues of the methodological approach is the absence of measurements through CROM in the three measurement times (CROM only used during the first experimentation). Since CROM is the method of reference and its reliability has been proved, why the authors did not perform CROM measurements in the two other times ? If the measurements were done, they have to be included. This aspect is essential since it defines the statistical method to be used for comparing both methods (IMU versus CROM).

Response 6: We think that the sequence objective of the study – study design – statistical analysis was not appropriately described in the text and this may, at least, in part, be the cause of the reviewer’s comments.

We aimed to identify the clinimetric properties, in terms of validity and reliability, of the craniocervical ROM evaluation in CP subjects and matched controls assessed by IMU. Therefore, although both properties are related between them, two different approaches should be applied to identify them, one for validity, and one more for reliability proposes. This design is common for determining the clinimetric properties of any evaluation tool, and can be considered as a strength of the current study, considering most of studies in this field of research, exclusively studying validity, or reliability, with a marked lack of studies that determine reliability values (Poitras et al., 2019).

A common pattern of the sequence objective of the study – study design – statistical analysis, similar to ours, can be identified in papers that  have tried to specifically determine both clinimetric properties for the application of IMU technology in health research (Al-Amri et al., 2018; Chang et al., 2019; Cho et al., 2018; Rantalainen, Gastin, Spangler, & Wundersitz, 2018; Rantalainen, Pirkola, Karavirta, Rantanen, & Linnamo, 2019; Yoon, Kim, & Min, 2019):

- First, for validity purposes, assessment of the reference method, for which reproducibility has been well established by previous research (Audette, Dumas, Côté, & De Serres, 2010; Fletcher & Bandy, 2008), makes it unnecessary to examine its reliability (Chang et al., 2019). This is compared with the first assessment of the method that is under study (Bauer et al., 2015);

- Second, for reliability purposes, regardless of whether the aim of the study was intra-day, inter-days, test-retest, intra-rater or inter-raters reliability,  the different assessments of the method that is under study are compared, including the first assessment, already used previously for validity comparisons (Bauer et al., 2015).

In summary, the first part of our study compared CROM data with IMU data, when both are assessed together, for concurrent validity purposes. Subsequently, a second part aimed to determine absolute and relative reliability features by comparing IMU vs IMU, both for intra-day and inter-day comparisons.

To clarify this in the paper, new sub-sections have been described in the Statistical Analysis, as done in the Results section, dividing the text in terms of Validity and Reliability. Furthermore, a comment has been added about the reliability of the three repetitions of each movement when the CROM was applied together with the IMU.

Lines 137-139: As the CROM was applied together with IMU, three repetitions of each movement were also performed, for which the ICC of the three repetitions was over 0.75 in all cases.

Point 7: In this case, the use of a t-test comparison is not appropriate. Why the authors did not use ANOVA (each patient or line representing a repeated measure) followed by a multiple comparison post-test ? If an unpaired t-test is used when comparing more than two groups, the p value obtained has to be divided by the number of groups for the obtention of a correct statistical threshold which is not the case in the current study.

Response 7: As commented in the response to Point 6, we have attempted to clarify the objective of each statistical analysis performed in the new version of the manuscript. Thus, as the study design has two different parts, data were analyzed separately.

First, for concurrent validity purposes, a paired t-test was used to compare CROM and IMU data when applied together (first evaluation of the first day).

Secondly, for reliability purposes, no direct comparison between means was performed in the first version of the paper. We have applied a two-way ANOVA according to the reviewer’s comments that can solve this issue (included as Supplementary material).

One more comparison was performed to identify possible differences between groups. Although this comparison was not completely specified in the first version of the paper, in the new version we explain that the between group comparisons included all ROM data: CROM, IMU first evaluation of the first day, IMU second evaluation of the first day, IMU evaluation of the second day. Unpaired t-tests were applied for this purpose.

All these clarifications were detailed along with the Statistical analysis, and some parts have been completely rewritten.

Although several analyses remain as presented in the original version of the paper, we agree with the reviewer that it may be interesting to perform an analysis with a better control of type I error risk, considering that some data are included in several comparisons, and no adjustment of the p-value threshold was included, based on the different aims of the study. One of the models that better fits this objective is the two-way ANOVA, with Evaluation (CROM, IMU first assessment on the first day; IMU second assessment on the first day; IMU assessment on the second day) as the within-subject factor, and Group (CP group; control group) as the between-subjects factor, which was performed for each spatial plane ROM. The evaluation-by-group interaction and both factors were of interest. In the case of a significant result, Bonferroni’s test was used for post-hoc analysis. The results of the ANOVA, its main factors, and the post-hoc analyses have now been added both in the text of the Results section and as Supplementary material (Table S1).

Lines 231-238: For a better control of type I error risk, due to the repeated comparison among CROM and IMU data, a two-way ANOVA, with Evaluation (CROM, IMU first assessment on the first day; IMU second assessment on the first day; IMU assessment on the second day) as the within-subject factor, and Group (CP group; control group) as the between-subjects factor, was performed for each spatial plane ROM. The evaluation-by-group interaction and both factors were of interest. Should the interaction or any of both factors reveal significance, the Bonferroni’s post-hoc test was used to verify whether a difference existed between the groups and/or within groups (view Supplementary Material, Table S1).

Results: 

Point 8: Some results are rather unusual. although the correlation between the two methods for measurement is particularly high (flexion-extension > 0.9 and side-bending > 0.9 for both groups), the values are significantly different when comparing both methods. This is not logical when, in addition to that, the SD represent almost 20% of the values. This issue questions the reliability and pertinence of the Student test for statistically comparing both groups (see my previous comment concerning the multiple comparison methods and the correction of statistical thresholds)?

Response 8: We are not sure why the reviewer thinks that these results are unusual. In statistical terms, it is not difficult to find very strong correlations (r>0.9), with statistical differences between data sets. On the other hand, as a measure of data dispersion, the standard deviations (SDs) of the means can influence the correlation between both variables, but it also depends on the covariance of the data sets. We have run the statistical analyses again, and descriptive data and correlations coefficients do not change. Similar patterns of results, with very strong correlations (r>0.9) and statistical differences between means, can be found in previous research with similar aims (Chang et al., 2019).

However, new statistical comparisons were performed to prevent possible misinterpretations due to multiple comparisons, as explained in the previous Point of this response letter.

Lines 266-274: No differences were detected between CP subjects and controls in each ROM for any of the assessments, regardless of the method of measurement (p>0.05).

Additionally, as reported in Table S1, the two-way ANOVA of the ROM of the three spatial planes showed a consistent pattern, with neither evaluation-by-group interaction nor Group factor significance, although the Evaluation factor detected statistical differences (p≤0.02). The post-hoc analysis of the Evaluation factor showed differences between CROM and the IMU assessments, with no differences among the three IMU assessments. The only exception to this pattern was the post-hoc analysis of the Evaluation factor concerning the rotational plane ROM, with statistical differences, exclusively, between the CROM and IMU assessments on the second day.

Point 9: Table 2 is rather unclear and could be simplified (e.g. Student-t value and exact p value and * are redundant; mean difference and 95%CI are difficult to read, what is the usefulness of these details?)

Response 9: We have simplified the Table 2 according to the reviewer’s recommendation.

Point 10: "LoAs" is not sufficiently explained nor declared in the methodology. It's relevance is not sufficiently commented.

Response 10: We thank the reviewer for noticing this omission. We have now included the relevance of LoAs, according to the literature.

Lines 207-210: The mean bias, defined as the average of the differences between both methods of measurement, was determined, together with limits of agreement (LoA), providing an estimate of the interval where 95% of the differences between both methods lie, and defined as the bias ±1.96 standard deviations of differences.

Additional explanations were included in the Discussion for a better interpretation of LoAs.

Lines 355-361: Although some methodological differences can be described between the research by Chang et al. and our own, such as the use of a sample comprising only healthy subjects, the assessment of movements which was performed from the neutral position, the use of a universal goniometer, and the absence of inter-day assessments, the interpretation of the agreement between both methods were the same. Thus, although the range between LoAs only defines the intervals of agreements, and not whether those limits are acceptable or not (Giavarina, 2015), we agree with Chang et al. that LoAs over 12º impaired the interchangeability of measurement methods (Chang et al., 2019).

Point 11: Figures 2 and 3 are difficult to interpret. It is said that craniocervical ranges are represented for IMU and CROM but few is explained inside the figures. Please explain better in the legend the meaning of these graphical representations.

Response 11: As recommended by the reviewer, we rewrote the legends of Figures 2 and 3. The new versions include how the plots were performed, originated with the IMU first assessment of the first day data and the CROM data. Additionally, some changes were introduced in both figures to improve comprehension.

Figure legend: Figure 2. Bland–Altman plots for craniocervical ranges of CP group measured by IMU first assessment on the first day and the CROM in (A) flexion-extension range, (B) rotational range, (C) side-bending range. The red line indicates the mean bias, whereas the blue lines refers to its upper and lower limits (mean ± 1.96 standard deviation). All plots were developed as follows: the Y axis corresponds to the differences between the paired values of both methods (IMU - CROM), whereas the X axis represents the respective value of the average of both (IMU + CROM / 2).

Figure legend: Figure 3. Bland–Altman plots for craniocervical ranges of Control group measured by IMU first assessment on the first day and the CROM in (A) flexion-extension range, (B) rotational range, (C) side-bending range. The red line indicates the mean bias, whereas the blue lines refers to its upper and lower limits (mean ± 1.96 standard deviation). All plots were developed as follows: the Y axis corresponds to the differences between the paired values of both methods (IMU - CROM), whereas the X axis represents the respective value of the average of both (IMU + CROM / 2).

Point 12: Concerning Intra-day and Inter-day Reliability, it is difficult to explain why the authors do not use as a control the CROM method, which is been proved to be reliable, in particular when comparisons between the two measurement methods at initial time showed a significant difference. In this sense, what is the rationale of the intra-day and inter-day comparisons solely through IMUs ?

Response 12: We respectfully disagree with the reviewer. We think it is important to comment that studies that aim of determine the clinimetric properties of assessment tools, commonly analyze those properties separately. An extensive justification of this statement can be found in our Response to Points 6 and 7.

We also think that the sequence objective of the study – study design – statistical analysis was not appropriately described in the first version of the manuscript. In the new version, that sequence was better explained, making it easier to understand which methods and assessments were selected for each study aim.

Point 13: Since the reliability of IMUs is only partially validated in the first part of the study, the methodology for the second part is not correct and should include CROM measurements. This will allow to use an adapted statistical method which will allow to provide the right statistical results for two-by-two comparisons (ANOVA for repeated measures followed by a multiple comparison post-test; see my previous comments).

Response 13: To respond to this Point, further information on the context was described in Response to Points 6 and 7. We think that the improvements introduced in the new version of the paper, which details the sequence objective of the study – study design – statistical analysis may help to a improve understanding of the text.

Specifically, since CROM is not the only reference method for determining ROM, and the possibility of a new assessment tool being reliable but not valid (or more reliable than valid, or as reliable as valid, or valid but not reliable), both clinimetric features were assessed separately, as is commonly performed in the literature and as explained in the Point 6 of this response letter.

However, for a better control of type I risk error, new analyses were performed by using mixed ANOVA models, according to the reviewer’s comments, and as explained in Point 7 of this response letter.

Lines 231-238: For a better control of type I error risk, due to the repeated comparison among CROM and IMU data, a two-way ANOVA, with Evaluation (CROM, IMU first assessment on the first day; IMU second assessment on the first day; IMU assessment on the second day) as the within-subject factor, and Group (CP group; control group) as the between-subjects factor, was performed for each spatial plane ROM. The evaluation-by-group interaction and both factors were of interest. Should the interaction or any of both factors reveal significance, the Bonferroni’s post-hoc test was used to verify whether a difference existed between the groups and/or within groups (view Supplementary Material, Table S1).

Lines 268-274: Additionally, as reported in Table S1, the two-way ANOVA of the ROM of the three spatial planes showed a consistent pattern, with neither evaluation-by-group interaction nor Group factor significance, although the Evaluation factor detected statistical differences (p≤0.02). The post-hoc analysis of the Evaluation factor showed differences between CROM and the IMU assessments, with no differences among the three IMU assessments. The only exception to this pattern was the post-hoc analysis of the Evaluation factor concerning the rotational plane ROM, with statistical differences, exclusively, between the CROM and IMU assessments on the second day.

Discussion:

Point 14: Since some methodological aspects are inadequately considered, the authors cannot affirm that the IMUs have a good concurrent validity regarding CROM and a very good intra-day and inter-day reliability (lines 266-274).

Response 14: We apologize for the errors and omissions identified in the methodology and thank the reviewer for their detailed considerations. New subsections, and more details and corrections, have been introduced throughout the entire Materials and Methods section. We have also inserted changes to the Results section, Tables and Figure legends to write all specific terminology in a more consistent manner.

Furthermore, almost all the new analyses performed, such as two-way ANOVA or the correlations between ROM and muscle tone (specific suggestion of Reviewer 2), agree with the previous analyses, which enabled us to introduce deeper interpretations, explanations and relevant references for a more focused discussion of the results.

Point 15: In line 285-286, the authors argue that the low correlation of CROM an IMU found in the rotational plane is due to the postural impairment of CP patients.

This can be overcome by using a moulded sit and back braces for example. A first step is to prove the interchangeability of both methods (CROM and IMU) and the reliability of IMU in optimal experimental conditions (i.e. without major biais) and then its use in day-to-day conditions (e.g. in a CP patient seated in a conventional chair).

Response 15: We agree with the reviewer. Regarding the lower correlations found in the rotational plane, this is a common result in research with IMUs, not only in CP patients but in healthy subjects, where lower validity and reliability results have been reported for the transverse/rotational plane (Poitras et al., 2019). New information was added at this point for a better interpretability of the results.

Lines 348-351: Indeed, the lower validity of the results concerning the rotational plane has been consistently reported in healthy adult subjects (Poitras et al., 2019), thus highlighting the significant challenge related to the measurement of ROM in rotational plane movements.

Lines 390-393: As commented, for validity purposes, the poorer results of the rotational plane have been also found in healthy adult subjects, including ICC values below 0.8 (Poitras et al., 2019). Further research and innovative assessment approaches are necessary to improve the quality of rotational plane ROM measurements.

Some other assessment protocols can be considered to assess craniocervical ROM in CP, and further research is necessary to adapt the assessment in any clinical and day-to-day conditions, which can increase the field of knowledge. Our protocol was designed to be applied in different motor function impairments (Gross Motor Function Classification System levels I to IV), due to the simplicity of the task performed (uniaxial movements) in a homogenous pattern (i.e.: A non-swivel chair was used, adapted to the anthropometric characteristics of each subject, who were seated in a standardized manner, and secured with straps when necessary. Specific instructions were given to the subject for the performance of each movement, as follows:…) and in a controlled setting.

Therefore, these other possibilities were added to the study limitations and proposed as future research lines.

Lines 447-453: The applicability of findings is limited to similar samples and assessment protocols. A wider scope is necessary to establish conclusions regarding specific GMFCS levels or other age ranges. Furthermore, the current study only assessed the ROM of simple movements in a specific and controlled setting, which limits the applicability of the results to more complex tasks and day-to-day conditions (Cuesta-Vargas, Galán-Mercant, & Williams, 2010). In fact, although more simple movements are used to produce better clinimetric properties (Walmsley et al., 2018), this approach did not solve the common measurement problems of rotational plane mobility (Poitras et al., 2019).

Lines 460-464: Further research is necessary, considering additional factors, such as other movement characteristics, including velocity, acceleration or coupled angles, and innovative assessment protocols, with a special focus on complex and day-to-day tasks and rotational plane movements, and larger sample sizes, in order to standardize technical procedures and obtain accurate and normative data (Poitras et al., 2019).

References cited in this review:

Al-Amri, M., Nicholas, K., Button, K., Sparkes, V., Sheeran, L., & Davies, J. L. (2018). Inertial measurement units for clinical movement analysis: Reliability and concurrent validity. Sensors (Switzerland), 18(3), 1–29. https://doi.org/10.3390/s18030719

Audette, I., Dumas, J. P., Côté, J. N., & De Serres, S. J. (2010). Validity and between-day reliability of the cervical range of motion (CROM) device. Journal of Orthopaedic and Sports Physical Therapy, 40(5), 318–323. https://doi.org/10.2519/jospt.2010.3180

Bauer, C. M., Rast, F. M., Ernst, M. J., Kool, J., Oetiker, S., Rissanen, S. M., … Kankaanpää, M. (2015). Concurrent validity and reliability of a novel wireless inertial measurement system to assess trunk movement. Journal of Electromyography and Kinesiology, 25(5), 782–790. https://doi.org/10.1016/j.jelekin.2015.06.001

Chang, K. V., Wu, W. T., Chen, M. C., Chiu, Y. C., Han, D. S., & Chen, C. C. (2019). Smartphone application with virtual reality goggles for the reliable and valid measurement of active craniocervical range of motion. Diagnostics, 9(3), E71. https://doi.org/10.3390/diagnostics9030071

Cho, Y. S., Jang, S. H., Cho, J. S., Kim, M. J., Lee, H. D., Lee, S. Y., & Moon, S. B. (2018). Evaluation of validity and reliability of inertial measurement unit-based gait analysis systems. Annals of Rehabilitation Medicine, 42(6), 872–883. https://doi.org/10.5535/arm.2018.42.6.872

Cuesta-Vargas, A. I., Galán-Mercant, A., & Williams, J. M. (2010). The use of inertial sensors system for human motion analysis. Phys Ther Rev, 15(6), 462–473.

Fletcher, J. P., & Bandy, W. D. (2008). Intrarater reliability of CROM measurement of cervical spine active range of motion in persons with and without neck pain. Journal of Orthopaedic and Sports Physical Therapy, 38(10), 640–645. https://doi.org/10.2519/jospt.2008.2680

Giavarina, D. (2015). Understanding Bland Altman analysis. Biochemia Medica, 25(2), 141–151. https://doi.org/10.11613/BM.2015.015

Poitras, I., Dupuis, F., Bielmann, M., Campeau-Lecours, A., Mercier, C., Bouyer, L. J., & Roy, J. S. (2019). Validity and reliability ofwearable sensors for joint angle estimation: A systematic review. Sensors (Switzerland), 19(7), E1555. https://doi.org/10.3390/s19071555

Rantalainen, T., Gastin, P. B., Spangler, R., & Wundersitz, D. (2018). Concurrent validity and reliability of torso-worn inertial measurement unit for jump power and height estimation. Journal of Sports Sciences, 36(17), 1937–1942. https://doi.org/10.1080/02640414.2018.1426974

Rantalainen, T., Pirkola, H., Karavirta, L., Rantanen, T., & Linnamo, V. (2019). Reliability and concurrent validity of spatiotemporal stride characteristics measured with an ankle-worn sensor among older individuals. Gait and Posture, 74(April), 33–39. https://doi.org/10.1016/j.gaitpost.2019.08.006

Walmsley, C. P., Williams, S. A., Grisbrook, T., Elliott, C., Imms, C., & Campbell, A. (2018). Measurement of Upper Limb Range of Motion Using Wearable Sensors: A Systematic Review. Sports Medicine - Open, 4(1), 53. https://doi.org/10.1186/s40798-018-0167-7

Yoon, T. L., Kim, H. N., & Min, J. H. (2019). Validity and Reliability of an Inertial Measurement Unit–based 3-Dimensional Angular Measurement of Cervical Range of Motion. Journal of Manipulative and Physiological Therapeutics, 42(1), 75–81. https://doi.org/10.1016/j.jmpt.2018.06.001

Reviewer 2 Report

This manuscript describes results from validity and reliability test of Inertial Measurement Units for assessment of craniocervical range or motion.  Controls and subjects with CP were evaluated twice. The results were compared with CROM for validation. Reliability was measured using ICC and MDC. There were good correlation on sagittal and frontal planes but not on transverse plane. Absolute reliability was variable.

The authors validated another important assessment tool, with a warning that it shouldn’t used to replace CROM.

I have the following questions/comments

Any reason for limiting the inclusion age to 14? During the assessment of children with CP, how is spasticity controlled? As it varies day to day, do you think that could be the reason on the variability in the absolute reliability? Can you include a correlation analysis of the Ashworth score with ROM? Can you also provide the variability of ROM in control subjects?  Please clarify in the IMU assessment section, whether the range of motion for each movement was kept consistent. If it wasn’t, how does it affect the readings? Due to the inflexibility of CROM assessment headband, foams were used. Based on physics of inertia, the foam is adding the radius of curvature (or moment arm). How is this accounted in the calculation? Based on your results, is CROM or IMU more affected by change in ROM due to spasticity? Can you discuss any advantage of using IMU rather than the validated CROM ? Also, can you comment on tolerability?

Author Response

Response to Reviewer 2 Comments

Point 1: This manuscript describes results from validity and reliability test of Inertial Measurement Units for assessment of craniocervical range or motion.  Controls and subjects with CP were evaluated twice. The results were compared with CROM for validation. Reliability was measured using ICC and MDC. There were good correlation on sagittal and frontal planes but not on transverse plane. Absolute reliability was variable.

The authors validated another important assessment tool, with a warning that it shouldn’t used to replace CROM.

Response 1: The authors appreciate the reviewer’s comments. We hope that the IMU can could be a useful assessment tool for the evaluation of craniocervical region in CP patients. In the current version of the manuscript, we provide further details about the methodology and how it was implemented, and new results have been included.

All changes made in the new version of the manuscript (diagnostics-694750_R1-WithFigures&Tables) are highlighted in red.

Point 2: I have the following questions/comments. Any reason for limiting the inclusion age to 14?

Response 2: This is an interesting point. We tried to ensure a homogenous assessment protocol, and other ages may have introduced too much data variability. We think that a 4-14 years old range is relevant for clinical purposes, although, as commented at the beginning of the paragraph on study limitations.

Lines 448-449: A wider scope is necessary to establish conclusions regarding specific GMFCS levels or other age ranges.

Furthermore, although pediatric care begins during infancy (between birth and 2 years of age) and can end in late adolescence (ages 17–21 years) (Hardin et al., 2017), other divisions can be found, and some Pediatric Services attend pediatric patients until early adolescence (ages 11–14 years), as occurs in Spain (Labay, 2012), the patients' country of origin.

Point 3: During the assessment of children with CP, how is spasticity controlled? As it varies day to day, do you think that could be the reason on the variability in the absolute reliability?

Response 3: We tried to apply simple tasks in a well pre-defined protocol, in order to control individual spasticity and, consequently, to reduce inter-assessment variability, as described in the Procedures sub-section.

Lines 155-160: The general recommendations for assessments in this patient profile were applied, meaning that evaluation and treatment strategies must include relatives or caregivers who are functionally involved and part of the daily relationship (relatives/caregiver/child) (Bartlett & Palisano, 2000, 2002).

The evaluations were performed in a quiet room, with no other people present besides the subject, assessors and relatives/caregiver. All people stood behind the study subject, except for the assessor who read the CROM values.

Lines 169-177: Manual stabilization was provided during each movement to avoid movements others than those requested and to control for any proprioceptive or other sensorimotor problems that could occur during the static posture or the performance of the movements, when necessary. To control for the appearance of resistance to movement due to spasticity, an assessor performed stretches of the muscle, repositioning the joint in the position where the resistance appeared. Subsequently, a second examiner annotated the CROM values (Van Den Noort, Scholtes, & Harlaar, 2009).

The Wong-Baker's facial pain scale (Bieri, Reeve, Champion, Addicoat, & Ziegler, 1990) was applied to assess whether patients suffered from pain throughout the evaluations. Its results were applied to interrupt the patient's participation in the study.

However, we have now added the possibility of inter-assessment variability being  due to spasticity as a possible source of absolute reliability, according to the reviewer’s comment.

Lines 403-408: Furthermore, specifically for inter-day calculations, two more sources of variability may explain these results. First, spasticity varies from one movement to another, and even more when the assessments are performed on different days (Bar-On et al., 2015), making it difficult to ensure that the evaluations of the CP subjects were performed in the same clinical conditions. Furthermore, it is known that spasticity can be influenced by apprehension, excitement and the position in which the child is assessed (Sarathy, Doshi, & Aroojis, 2019), which can increase the variability of the ROM results, mainly in inter-day evaluations. Second, the magnitude of a training effect or compensation cannot be calculated due to the repetitions (Chang et al., 2019), however, all the mean ROMs on the second day were higher than those of the first day in both groups, which may have also influenced the absolute reliability between days.

Point 4: Can you include a correlation analysis of the Ashworth score with ROM?

Response 4: We thank the reviewer for their interesting suggestion. We have added a correlation analysis between the Ashworth score and ROM in CP subjects.

Lines 195-198: Spearman's rho correlation coefficient (rs) was used to identify associations between cervical muscle tone and ROM, assessed with the CROM and IMU. Correlation coefficient values were considered poor when values were below 0.20, fair for values between 0.21 to 0.50, moderate from 0.51 to 0.70, very strong from 0.71 to 0.90 and almost perfect from 0.91 to 1.00 (Akoglu, 2018).

The main results showed that flexor, extensor and sternocleidomastoid muscles of CP subjects correlated exclusively with rotational ROM, with higher values of tone being associated with lower ROM, and vice versa.

Lines 257-265: The correlation analysis between MAS and ROM, assessed by the CROM and the IMU, showed a common trend, with flexor, extensor and sternocleidomastoid muscles of CP subjects, significantly and negatively correlated with rotational ROM (in all cases: the higher the muscle tone, the lower the ROM). Thus, the tone of flexor and extensor muscles correlated with: CROM: rs = -0.504; IMU first assessment on the first day: rs = -0.510; IMU second assessment on the first day: rs = -0.483; IMU assessment on the second day: rs = -0.412. Right and left sternocleidomastoid muscle tone correlated with: CROM: rs = -0.433; IMU first assessment on the first day: rs = -0.437; IMU second assessment on the first day: rs = -0.420; IMU assessment on the second day: rs = -0.410. No correlation was identified in the planes of flexion-extension and side-bending.

Further interpretations of these results were added to the Discussion.

Lines 404-408: First, spasticity varies from one movement to another, and even more when the assessments are performed on different days (Bar-On et al., 2015), making it difficult to ensure that the evaluations of the CP subjects were performed in the same clinical conditions. Furthermore, it is known that spasticity can be influenced by apprehension, excitement and the position in which the child is assessed (Sarathy et al., 2019), which can increase the variability of the ROM results, mainly in inter-day evaluations. Second, the magnitude of a training effect or compensation cannot be calculated due to the repetitions (Chang et al., 2019), however, all the mean ROMs on the second day were higher than those of the first day in both groups, which may have also influenced the absolute reliability between days.

Point 5: Can you also provide the variability of ROM in control subjects? 

Response 5: We are not certain what the reviewer means by this statement. The variability of ROM in both CP subjects and controls are included by the standard deviation values of the mean in Tables 2 and 3. The Mean difference (95% CI) between CROM and IMU of Table 2 was removed in the current version according to a specific comment by Reviewer 1.

Point 6: Please clarify in the IMU assessment section, whether the range of motion for each movement was kept consistent. If it wasn’t, how does it affect the readings?

Response 6: The variability of the results, in terms of velocity, accelerations, or coupled angles, was not analyzed in this study, as it was not the aim of the present study. We focused on the clinimetric properties of IMUs to assess ROM, based on the fact that this is a common assessment in CP patients. Nevertheless, other characteristics of mobility are interesting, and can be addressed in future research. As a result, a new limitation has been added.

Line 449-451: Furthermore, the current study only assessed the ROM of simple movements in a specific and controlled setting, which limits the applicability of the results to more complex tasks and day-to-day conditions (Cuesta-Vargas, Galán-Mercant, & Williams, 2010).

Line 460-464: Further research is necessary, considering additional factors, such as other movement characteristics, including velocity, acceleration or coupled angles, and innovative assessment protocols, with a special focus on complex and day-to-day tasks and rotational plane movements, and larger sample sizes, in order to standardize technical procedures and obtain accurate and normative data (Poitras et al., 2019).

Furthermore, the reliability of the three repetitions for each movement was added in the IMU assessment section.

Lines 124-125: The ICC among the three repetitions of each movement was over 0.8 in all cases.

Point 7: Due to the inflexibility of CROM assessment headband, foams were used. Based on physics of inertia, the foam is adding the radius of curvature (or moment arm). How is this accounted in the calculation?

Response 7: We agree with the reviewer, although we think that this possible source of error is small in the final results. Indeed, we did not find any research that takes this margin of error into account. Furthermore, this artefact can exclusively affect the reference method (CROM), but not the principal assessment method under study (IMU), which is the main aim of this research.

We rewrote a part of the Discussion section to include some comments about the applicability of CROM and IMU, including a comment on semi-rigid foams.

Lines 370-375: Analogic goniometry using CROM has been previously established as the method of reference for evaluating neck motion (Audette, Dumas, Côté, & De Serres, 2010; Fletcher & Bandy, 2008), however, recently, other 3-D kinematic devices have been also been proposed (Song et al., 2018), and some circumstances support the use of digital devices to assess neck motion in CP, such as the need of one less assessor to obtain the CROM data, a proper adjustment to the shape and size of the head without the need for additional elements (i.e. semi-rigid foams), for use in the pediatric population, and the elimination of the reading error associated to analogic devices (Chang et al., 2019; Choi, Shin, Kim, & Kim, 2018).

Point 8: Based on your results, is CROM or IMU more affected by change in ROM due to spasticity?

Response 8: As previously commented in Point 4 of this Response, a correlation analysis between the Ashworth scale and ROM was performed. We found a similar pattern of correlations, regardless of the use of CROM or IMU, and even regardless of the evaluation. This means that none of these two methods were more affected than the other due to spasticity.

The following statements have been added to the text:

Lines 257-265: The correlation analysis between MAS and ROM, assessed by the CROM and the IMU, showed a common trend, with flexor, extensor and sternocleidomastoid muscles of CP subjects, significantly and negatively correlated with rotational ROM (in all cases: the higher the muscle tone, the lower the ROM). Thus, the tone of flexor and extensor muscles correlated with: CROM: rs = -0.504; IMU first assessment on the first day: rs = -0.510; IMU second assessment on the first day: rs = -0.483; IMU assessment on the second day: rs = -0.412. Right and left sternocleidomastoid muscle tone correlated with: CROM: rs = -0.433; IMU first assessment on the first day: rs = -0.437; IMU second assessment on the first day: rs = -0.420; IMU assessment on the second day: rs = -0.410. No correlation was identified in the planes of flexion-extension and side-bending.

Lines 426-429: Indeed, regardless of the method of measurement, only fair to moderate correlations were found, exclusively between the tone of cervical muscles and rotational ROM, perhaps due to the fact that greater motor control is necessary to perform rotations, as previously described, with no correlations in any other spatial plane.

Point 9: Can you discuss any advantage of using IMU rather than the validated CROM? Also, can you comment on tolerability? 

Response 9: The advantages of using IMU rather than CROM for the craniocervical ROM assessment have now been updated in the Discussion section-

Lines 370-375: Analogic goniometry using CROM has been previously established as the method of reference for evaluating neck motion (Audette et al., 2010; Fletcher & Bandy, 2008), however, recently, other 3-D kinematic devices have been also been proposed (Song et al., 2018), and some circumstances support the use of digital devices to assess neck motion, such as the need of one less assessor to obtain the CROM data, a proper adjustment to the shape and size of the head without the need for additional elements (i.e. semi-rigid foams), for use in the pediatric population, and the elimination of the reading error associated to analogic devices (Chang et al., 2019; Choi et al., 2018).

Thus, a specific recommendation for using the IMU for  ROM assessment has been included in the Discussion, supported by recent literature.

Lines 377-379: Furthermore, a recent systematic review has suggested that rehabilitation research and health care services could benefit from IMUs because they provide valid data to assess ROM and joint orientation (Poitras et al., 2019).

Regarding tolerability, both methods were well accepted by the individuals of both study groups, and no subject stated any advantages compared to the other measurement method. No pain arose which may have lead to the interruption of the assessments, and therefore both methods are considered safe and innocuous.

Lines 417-419: No pain was experienced by the study subjects during the procedures, and no assessment was interrupted due to the evaluation protocol. This means that the application of IMUs for craniocervical ROM assessment is tolerable, safe and innocuous when applied to CP and healthy children.

References cited in this review:

Akoglu, H. (2018). User’s guide to correlation coefficients. Turkish Journal of Emergency Medicine, 18(3), 91–93. https://doi.org/10.1016/j.tjem.2018.08.001

Audette, I., Dumas, J. P., Côté, J. N., & De Serres, S. J. (2010). Validity and between-day reliability of the cervical range of motion (CROM) device. Journal of Orthopaedic and Sports Physical Therapy, 40(5), 318–323. https://doi.org/10.2519/jospt.2010.3180

Bar-On, L., Molenaers, G., Aertbeliën, E., Van Campenhout, A., Feys, H., Nuttin, B., & Desloovere, K. (2015). Spasticity and its contribution to hypertonia in cerebral palsy. BioMed Research International, 2015, 317047. https://doi.org/10.1155/2015/317047

Bartlett, D. J., & Palisano, R. J. (2000). A multivariate model of determinants of motor change for children with cerebral palsy. 80(6), 598–614.

Bartlett, D. J., & Palisano, R. J. (2002). Factors influencing the acquisition of motor abilities of children with cerebral palsy: implications for clinical reasoning. 82(3), 237–248.

Bieri, D., Reeve, R. A., Champion, D., Addicoat, L., & Ziegler, J. B. (1990). The Faces Pain Scale for the self-assessment of the severity of pain experienced by children: development, initial validation, and preliminary investigation for ratio scale properties. 41, 139–150.

Chang, K. V., Wu, W. T., Chen, M. C., Chiu, Y. C., Han, D. S., & Chen, C. C. (2019). Smartphone application with virtual reality goggles for the reliable and valid measurement of active craniocervical range of motion. Diagnostics, 9(3), E71. https://doi.org/10.3390/diagnostics9030071

Choi, S., Shin, Y. B., Kim, S. Y., & Kim, J. (2018). A novel sensor-based assessment of lower limb spasticity in children with cerebral palsy. Journal of NeuroEngineering and Rehabilitation, 15(1), 45. https://doi.org/10.1186/s12984-018-0388-5

Cuesta-Vargas, A. I., Galán-Mercant, A., & Williams, J. M. (2010, December 1). The use of inertial sensors system for human motion analysis. Physical Therapy Reviews, Vol. 15, pp. 462–473. https://doi.org/10.1179/1743288X11Y.0000000006

Fletcher, J. P., & Bandy, W. D. (2008). Intrarater reliability of CROM measurement of cervical spine active range of motion in persons with and without neck pain. Journal of Orthopaedic and Sports Physical Therapy, 38(10), 640–645. https://doi.org/10.2519/jospt.2008.2680

Hardin, A. P., Hackell, J. M., Simon, G. R., Boudreau, A. D. A., Baker, C. N., Barden, G. A., … Sobczyk, E. (2017). Age limit of pediatrics. Pediatrics, 140(3). https://doi.org/10.1542/peds.2017-2151

Labay, M. (2012). Paediatrics, the People and Politicians in Spain – History, Development, Reality and Future. In Contemporary Pediatrics (pp. 1–24). https://doi.org/10.5772/33225

Poitras, I., Dupuis, F., Bielmann, M., Campeau-Lecours, A., Mercier, C., Bouyer, L. J., & Roy, J. S. (2019). Validity and reliability ofwearable sensors for joint angle estimation: A systematic review. Sensors (Switzerland), 19(7), E1555. https://doi.org/10.3390/s19071555

Sarathy, K., Doshi, C., & Aroojis, A. (2019). Clinical examination of children with cerebral palsy. Indian Journal of Orthopaedics, 53(1), 35–44. https://doi.org/10.4103/ortho.IJOrtho_409_17

Song, H., Zhai, X., Gao, Z., Lu, T., Tian, Q., Li, H., & He, X. (2018). Reliability and validity of a Coda Motion 3-D Analysis system for measuring cervical range of motion in healthy subjects. Journal of Electromyography and Kinesiology, 38(157), 56–66. https://doi.org/10.1016/j.jelekin.2017.11.008

Van Den Noort, J. C., Scholtes, V. A., & Harlaar, J. (2009). Evaluation of clinical spasticity assessment in Cerebral palsy using inertial sensors. Gait and Posture, 30(2), 138–143. https://doi.org/10.1016/j.gaitpost.2009.05.011

Round 2

Reviewer 1 Report

All the identified methodological and scientific issues have been satisfactorily improved.